# OpenReview forum: "Measuring bias in Instruction-Following models with P-AT"
_EMNLP/2023/Conference — EMNLP 2023 Findings_

### Official Review · Reviewer_Ckmh · 2023-08-03

**Soundness:** 4

**Excitement:**

3: Ambivalent: It has merits (e.g., it reports state-of-the-art results, the idea is nice), but there are key weaknesses (e.g., it describes incremental work), and it can significantly benefit from another round of revision. However, I won't object to accepting it if my co-reviewers champion it.

**Missing References:**

Line 159: There are also issues with these metrics, since the tests contain sentences that are acually not encoding any biases, see: Stereotyping Norwegian Salmon: An Inventory of Pitfalls in Fairness Benchmark Datasets (Blodgett et al., ACL-IJCNLP 2021)

Other references:
[1] Bad Seeds: Evaluating Lexical Methods for Bias Measurement (Antoniak & Mimno, ACL-IJCNLP 2021)
[2] Measuring Fairness with Biased Rulers: A Comparative Study on Bias Metrics for Pre-trained Language Models (Delobelle et al., NAACL 2022)

**Paper Topic And Main Contributions:**

This paper presents a fairness metrics for instruction-tuned language models, as the current metrics do not fully translate to this kind of models. The metric is based on the WEAT seed words and some own instruction prompts as templates. Additionally, the paper also introduces the use of entropy for measuring the quality of predictions, which is a (to me) novel idea that makes a lot of sense.

The evaluation is done on a range of instruction-tuned language models and seems solid enough.

**Questions For The Authors:**

QA: One thing that's missing is the direction of bias. The tests (e.g. Age) are explained well, but the direction is not explained. For instance, Age on Flan-xxl is negative, does that mean this is the only model favoring younger or older people?
QB: How are the prompts found? Is there some strategy to it? Please specify this, as language models are extremely brittle and different prompts and seeds could lead to vastly different scores (cf. [1, 2])
Q3: Will code be made publicly available under a permissive license?

**Reasons To Accept:**

The authors identified a gap in the current literature on measuring biases in language models and presented a well-motived metric to measure these biases in instruction-tuned models. This is an important issue and the contribution is timely.

The evaluation is quite solid, although I have some issues with the presentation and stability of the results. Nevertheless, I am confident I could (at least in part) replicate some results.

**Reasons To Reject:**

There is some information missing in the current manuscript that makes understanding the published scores more difficult. I've added these items to the Questions, which I hope will be clarified and added to the manuscript.

What I'm also missing is a clearer bias description (which relates a bit to QA). What are the biases you are measuring precisely and how does a bad score affect ... something... What is the consequence of having a p-AT score of 1.0? I don't know if you need to do any experiments on this, but at least linking the score to the social issues you are trying to measure would be helpful.

Finally, one thing that concerns me is that some prompts give completely different results and some just gives values of 0.0. While I appreciate the openness of including these subscores, they do highlight the brittleness of the prompts. One question (QB) is how these prompts where obtained, but perhaps there needs to be more work done on making/finding less brittle prompts? An alternative could be reporting some more statistics.

**Reproducibility:**

3: Could reproduce the results with some difficulty. The settings of parameters are underspecified or subjectively determined; the training/evaluation data are not widely available.

**Reviewer Confidence:**

5: Positive that my evaluation is correct. I read the paper very carefully and I am very familiar with related work.

**Typos Grammar Style And Presentation Improvements:**

- Table 4 could be integrated in Table 3, no need to waste space on this.
- Table 3 is interesting but quite hard to parse. Is there a way of doing a statistical test to highlight statistically significant biased sets? Especially when the entropy is low I don't think the results mean anything useful.

---

> ### Author Rebuttal · Authors · 2023-08-28
>
> **Addressed Issue: Bias Definition, commenting Reasons to Reject**
>
> We can adapt the definition of bias that is given for Language Models in general to this scenario (that we adopt, among the others, from "StereoSet: Measuring stereotypical bias in pre-trained language models", Nadeem et al., 2019).
>
> In LMs, given a target, identifying a certain social group, LMs are tested by observing which contexts they prefer for each target among stereotyped and anti-stereotyped contexts: they are stereotypically biased if they systematically choose the stereotyped context given a target group.
>
> In IFLMs, we can use the previous definition, identifying both the context given to a model and a target word referring to a certain social group when formulating the question.
> The prompt, hence, consists of a classification task, and IFLMs are forced to answer by producing either a stereotype or an anti-stereotype answer.
> The strong stereotyped biased behavior is manifested in a model's tendency to answer by producing stereotyped associations more often than anti-stereotyped ones and results in a P-AT Bias Score close to 1.
> We will surely add these definitions.
>
> **Addressed Issue: Brittleness of the Prompts, commenting Reasons to Reject**
>
> The models are extremely sensitive to variations in the prompt, and at the moment, the choice we have made is to increase the number of tests to get meaningful results. At present, we have proposed 5 of them for semantically equivalent question, taking care to use language that is more or less formal and rephrasing the same semantic content with different syntax.
>
> **Addressed Issue: Direction of Bias, answering Question to the Author QA**
>
> The definition of stereotype bias refers to ingrained human biases and is identified with a positive value in the P-AT Bias Score. A negative score, on the other hand, indicates that a model tends to produce anti-stereotypes.
> As a borderline case, a score close to -1 means that a model is biased --that is, it tends to show an association toward a social group and a set of attributes-- but tends to produce anti-stereotype associations.
> However, in our experiments, negative results are also coupled with relatively low entropy, meaning that a model tends to ignore prompt variation.
>
> **Addressed Issue: Prompts Definition, answering Question to the Author QB**
>
> We absolutely agree that variation in prompts can lead to different results. Therefore, we collected fifteen prompts for each PAT task. Then, three annotators had a majority vote on whether a prompt was to be kept or discarded. The rationale was to include equivalent questions by varying the language register and syntax. The process was iterated until five prompts per task were selected.
>
> **Addressed Issue: License, answering Question to the Author Q3**
>
> Surely, we plan to publish both the code and the data under a Permissive license.

---

### Official Review · Reviewer_egsY · 2023-08-06

**Soundness:** 3

**Excitement:**

2: Mediocre: This paper makes marginal contributions (vs non-contemporaneous work), so I would rather not see it in the conference.

**Paper Topic And Main Contributions:**

This paper introduces P-AT, a resource for detecting race, gender, and age biases in instruction-following language models (IFLMs). P_AT is based on WEAT (Word Embedding Association Test), which is in turn based on the Implicit Association Test. The paper finds that off-the-shelf models like Alpaca and FLAN-T5-xxl show racial and gender biases on this test.

**Reasons To Accept:**

* proposes a bias detection resource specifically for IFLMs.
* provides experimental results for 6 off-the-shelf IFLMs, finding evidence of bias in most.

**Reasons To Reject:**

This paper has several key weaknesses:
* the paper's only contributions are rewriting WEAT as prompts for IFLMs and providing some baseline results on P-AT. The authors did not create a new dataset; instead they reformatted an existing dataset. This is a small, focused contribution and might be better suited for a short paper or workshop paper.
* The paper does not explain the decision to use WEAT as the basis for P-AT. There are many bias detection datasets that are more recent and more comprehensive than WEAT, and the paper needs more explanation for this design decision.
* I do not think the WEAT framework is an appropriate way to probe for bias in IFLMs. Comparing the distance between "science" and either "male" or "female" makes sense in the context of word embedding vectors. However, the prompts in P-AT instruct the models to produce stereotypes and make artificial judgements, which is not necessarily reflective of downstream use cases.
* The paper does not provide concrete definitions for "bias" and "harm". What harms does this work aim to prevent? What risks are specific to IFLMs and not inherent in the underlying models?

**Reproducibility:**

4: Could mostly reproduce the results, but there may be some variation because of sample variance or minor variations in their interpretation of the protocol or method.

**Reviewer Confidence:**

4: Quite sure. I tried to check the important points carefully. It's unlikely, though conceivable, that I missed something that should affect my ratings.

**Typos Grammar Style And Presentation Improvements:**

* Sec 2: this part is extremely long. Could be cut down.
* L184: should this say "cause no harm" or "cause harm"? sentence is unclear
* Sec 3.1: overview of WEAT could either be shortened or moved to an appendix
* Table 3: I don't fully understand the intended takeaway from the entropy values. I think this could be explained better.
* L513: should this be convex, not concave? if bias increases then decreases?

---

> ### Author Rebuttal · Authors · 2023-08-28
>
> **Addressed Issue: decision to use WEAT**
>
> P-AT is based on WEAT because it imitates the human Implicit Association Test (IAT) (Greenwald et al., 1998). This psychological study guarantees a correct association between a social case word and the stereotype or anti-stereotype.
>
> In addition to WEAT, newer and larger benchmark datasets such as StereoSet (SS), CrowS-Pairs (CS), WinoBias (WB), or Winogender (WG) are available for evaluating the bias of a model. However, in this regard, the reviewer **Ckmh** suggests that these large and new datasets have limitations (Blodgett et al., ACL-IJCNLP 2021).
>
> Therefore, this decision design limits the dataset size, but it guarantees the reliability of the association test.
>
> **Addressed Issue: Prompts in P-AT and downstream tasks**
>
> Judgments required by prompts in P-AT are not so far from possible downstream usage. For example, a user can easily feed a resume or a motivation letter to get help analyzing these documents. Then, s/he may ask: "Is John Brown reliable?". In P-AT Race, we demonstrate that even the name John can positively or negatively influence the answer the specific prompt. The same holds when they are asked to judge whether someone is inclined to study in scientific areas or artistic ones: PAT-Base subtasks and PAT-gender demonstrate that these models tend to associate the scientific and mathematical fields of study with men and the artistic field with women.
> We totally agree that P-AT is not the only possible bias test but it is a possible solution.
>
> **Addressed Issue: Bias and Harm definition**
>
> We consider a model to be stereotype-biased if it consistently prefers stereotypes over anti-stereotypes; in this scenario we say that the model exhibits stereotypical bias. In a IFLM, the strong stereotyped biased behavior is manifested when the outputs produced by the model have stereotyped associations more often than anti-stereotyped ones.
>
> IFLMs can have a negative impact on the society, for example many recruiting platforms filter the applications through AI models. If these filters favor a stereotype a priori, they are a serious problem. With the same curricula, a model with a bias filter can prefers a candidate with a European name instead of African. Similarly, certain gender-stereotyped attributes have been strongly associated with occupational salary and prestige (Glick, 1991).

---

### Official Review · Reviewer_UNmW · 2023-08-07

**Soundness:** 4

**Excitement:**

4: Strong: This paper deepens the understanding of some phenomenon or lowers the barriers to an existing research direction.

**Paper Topic And Main Contributions:**

* The study focuses on extending the WEAT task for Instruction-Following Language Models (IFLMs) like Alpaca and Flan-T5.
* P-AT adapts Word Embedding Association Test (WEAT) to prompt-based format for IFLMs, enabling the detection of biases across dimensions like gender, race, and age.
* In addition to the bias score, an evaluation metric called Entropy is introduced to assess the language modeling performance of the system. While the bias score measures alignment with human biases, entropy evaluates whether the model is making meaningful choices between responses.
* The authors apply their bias assessment to several state-of-the-art models, revealing biases towards various sensitive attributes such as gender and race.
* The study also highlights an intriguing relationship between bias and model sizes, along with demonstrating the significant impact of prompt templates on model bias.

**Questions For The Authors:**

1. How would you ensure that the bias in these models are significant and aren't due to the chance? The problem of considering a biased model as non-biased is more likely in your work compared to SEAT/WEAT
2. I see that sometimes changing the order of attributes in inputted prompt can change the answer of the model. Did you consider that in your variation of prompts experiment?

**Reasons To Accept:**

This paper presents a highly valuable tool for assessing bias within models capable of comprehending prompts. The study's approach is presented in a clear and straightforward manner, supported by well-defined experiments and descriptions. While WEAT metric might not be the most optimal choice for evaluating bias in language models, its significance lies in its extensive history of use and study, makes it a valuable inclusion in the prompt template. The introduction of the Entropy metric is a commendable addition, aiding in the distinction between low bias scores resulting from the model's contextual understanding limitations and those arising from genuinely reduced bias encoding.

**Reasons To Reject:**

There is a key weakness in P-AT compared to prior work like SEAT/WEAT - it does not directly measure the statistical significance of the observed bias. SEAT and WEAT compute a p-value using permutation testing to quantify whether the measured association is significant or likely due to chance. However, calculating p-values is very challenging in a prompt-based setting like P-AT. Since we only observe the model's generated text rather than its internal statistical representations, we cannot easily construct a permutation test. P-AT tried to compensate by introducing an entropy measure, which indicates whether the model produces varied responses to the prompts. But entropy does not actually test the significance of the bias score. At best, higher entropy provides some evidence that the model is not ignoring the prompts entirely. The entropy metric in P-AT is still useful - it captures whether the model is a good language model in this context. But it does not directly substitute for a significance test like the p-value in SEAT and WEAT.

As previously mentioned, it's worth noting that WEAT may not serve as the most optimal bias metric for evaluating bias due to several inherent weaknesses. The rest of my criticisms directed towards this research can similarly be extended to WEAT.

Initially, WEAT's reliance on templates raises concerns about its applicability, especially concerning new large language models renowned for their advanced capabilities. Additionally, the operational mechanism of the WEAT bias metric has the potential to obscure bias in certain edge cases. For instance, in cases where a model displays an equivalent frequency of stereotyped and anti-stereotyped behaviors (wherein both scenarios the model exhibits biased behavior towards some group to some extent), the opposing positive and negative signs might offset each other in the bias formula, creating an illusion of an unbiased model despite clear underlying bias.

**Reproducibility:**

4: Could mostly reproduce the results, but there may be some variation because of sample variance or minor variations in their interpretation of the protocol or method.

**Reviewer Confidence:**

4: Quite sure. I tried to check the important points carefully. It's unlikely, though conceivable, that I missed something that should affect my ratings.

---

> ### Author Rebuttal · Authors · 2023-08-28
>
> **Addressed Issue: Statistical Significance of Results, commenting reasons to reject and first question**
>
> *Short answer*: We can definitely introduce a test to assess the significance of the results! In particular, we can rely on Fisher's exact test for contingency tables.
>
> *Details*: The test aims to examine the significance of the association between the two kinds of classification for categorical data.
> In P-AT we observe the occurrences of Attributes with respect to the social groups (Targets): we want to test whether any difference in proportions that we observe is significant.
> The null hypothesis states the independence of the two categorical variables Targets and Attributes or, in other words, that the observed differences in proportion are only due to chance.
> Under the null hyphotesis, the numbers in the cells of the table have a hypergeometric distribution. A low p-value under a certain alpha (that we fix at 0.05 and 0.10) means that the null hyphotesis can be rejected, and hence we state the significance of the results.
>
> In the table below -- an update of Table 3 in the paper -- we mark statistically significant results with a p-value lower than 0.10 as \*  and with \*\* the results significant with a p-value lower than 0.05.
>
> | PAT-Task       | WEAT-task | model   | score | entropy |
> |--------------------|---------------|---------------|-----------|-------------|
> | P-AT-base	     | weat1         | vicuna-7b     | 0.56\*\*    | 0.86        |
> |                    | weat2         | vicuna-7b     | 0.15\*\*    | 0.73        |
> |                    | weat3         | vicuna-7b     | -0.0      | 0.14        |
> |                    | weat3b        | vicuna-7b     | -0.08     | 0.36        |
> |                    | weat4         | vicuna-7b     | 0.02      | 0.08        |
> |                    | weat10        | vicuna-7b     | 0.0       | 0.0         |
> |	             | weat6         | vicuna-7b     | -0.01     | 0.0         |
> |		     | weat7         | vicuna-7b     | 0.24\*\*    | 0.33        |
> |                    | weat8         | vicuna-7b     | 0.15      | 0.53        |
> |		     | weat9         | vicuna-7b     | -0.11     | 0.4         |
> | P-AT-race          | weat3         | vicuna-7b     | 0.06      | 0.22        |
> |                    | weat4         | vicuna-7b     | 0.04      | 0.27        |
> | P-AT-gender        | weat7         | vicuna-7b     | 0.0       | 0.0         |
> |                    | weat8         | vicuna-7b     | 0.02      | 0.14        |
> |                    | weat6         | vicuna-7b     | 0.02      | 0.3         |
> | P-AT-age           | weat10        | vicuna-7b     | -0.12     | 0.18        |
> | P-AT-base	    | weat1         | llama7b       | 0.72\*\*    | 0.97        |
> |                    | weat2         | llama7b       | 0.47\*\*    | 0.9         |
> |                    | weat3         | llama7b       | 0.27\*\*    | 0.52        |
> |                    | weat3b        | llama7b       | 0.17\*\*    | 0.48        |
> |                    | weat4         | llama7b       | 0.18\*\*    | 0.32        |
> |                    | weat10        | llama7b       | 0.16\*     | 0.44        |
> |                        | weat6         | llama7b       | 0.15\*\*    | 0.33        |
> |                        | weat7         | llama7b       | 0.41\*\*    | 0.55        |
> |                    | weat8         | llama7b       | 0.39\*\*    | 0.54        |
> |                       | weat9         | llama7b       | 0.13      | 0.57        |
> | P-AT-race          | weat3         | llama7b       | 0.67\*\*    | 0.91        |
> |                    | weat4         | llama7b       | 0.61\*\*    | 0.93        |
> | P-AT-gender        | weat7         | llama7b       | 0.4\*\*     | 0.53        |
> |                    | weat8         | llama7b       | 0.35\*\*    | 0.56        |
> |                    | weat6         | llama7b       | 0.15\*\*    | 0.34        |
> | P-AT-age           | weat10        | llama7b       | 0.2       | 0.89        |
> | P-AT-base | weat1         | flan-t5-base  | 0.39\*\*    | 0.64        |
> |                    | weat2         | flan-t5-base  | 0.28\*\*    | 0.48        |
> |                    | weat3         | flan-t5-base  | 0.14\*\*    | 0.49        |
> |                    | weat3b        | flan-t5-base  | 0.11      | 0.33        |
> |                    | weat4         | flan-t5-base  | 0.08      | 0.62        |
> |                    | weat10        | flan-t5-base  | 0.15      | 0.46        |
> |                        | weat6         | flan-t5-base  | -0.1      | 0.51        |
> |                        | weat7         | flan-t5-base  | 0.18      | 0.29        |
> |                    | weat8         | flan-t5-base  | 0.15      | 0.18        |
> |                       | weat9         | flan-t5-base  | -0.2      | 0.46        |
> | P-AT-race          | weat3         | flan-t5-base  | 0.26\*\*    | 0.39        |
> |                    | weat4         | flan-t5-base  | 0.17\*\*    | 0.32        |
> | P-AT-gender        | weat7         | flan-t5-base  | 0.4\*\*     | 0.63        |
> |                    | weat8         | flan-t5-base  | 0.28\*\*    | 0.65        |
> |                    | weat6         | flan-t5-base  | 0.04      | 0.11        |
> | P-AT-age           | weat10        | flan-t5-base  | 0.12      | 0.2         |
> | P-AT-base | weat1         | flan-t5-large | 0.58\*\*    | 0.92        |
> |                    | weat2         | flan-t5-large | 0.65\*\*    | 0.88        |
> |                    | weat3         | flan-t5-large | 0.2\*\*     | 0.62        |
> |                    | weat3b        | flan-t5-large | 0.0       | 0.46        |
> |                    | weat4         | flan-t5-large | 0.11      | 0.43        |
> |                    | weat10        | flan-t5-large | 0.15      | 0.44        |
> |                        | weat6         | flan-t5-large | 0.08      | 0.31        |
> |                        | weat7         | flan-t5-large | 0.49\*\*    | 0.81        |
> |                    | weat8         | flan-t5-large | 0.5\*\*     | 0.86        |
> |                       | weat9         | flan-t5-large | 0.17      | 0.37        |
> | P-AT-race          | weat3         | flan-t5-large | 0.03      | 0.25        |
> |                    | weat4         | flan-t5-large | 0.09      | 0.25        |
> | P-AT-gender        | weat7         | flan-t5-large | 0.42\*\*    | 0.68        |
> |                    | weat8         | flan-t5-large | 0.35\*\*    | 0.73        |
> |                    | weat6         | flan-t5-large | 0.05      | 0.25        |
> | P-AT-age           | weat10        | flan-t5-large | 0.05      | 0.73        |
> | P-AT-base | weat1         | flan-t5-xl    | 0.8\*\*     | 0.99        |
> |                    | weat2         | flan-t5-xl    | 0.7\*\*     | 0.99        |
> |                    | weat3         | flan-t5-xl    | 0.22\*\*    | 0.49        |
> |                    | weat3b        | flan-t5-xl    | 0.09      | 0.25        |
> |                    | weat4         | flan-t5-xl    | 0.2\*\*     | 0.54        |
> |                    | weat10        | flan-t5-xl    | 0.2\*\*     | 0.4         |
> |                        | weat6         | flan-t5-xl    | 0.3\*\*     | 0.7         |
> |                        | weat7         | flan-t5-xl    | 0.87\*\*    | 0.99        |
> |                    | weat8         | flan-t5-xl    | 0.7\*\*     | 0.98        |
> |                       | weat9         | flan-t5-xl    | 0.17      | 0.91        |
> | P-AT-race          | weat3         | flan-t5-xl    | 0.12\*\*    | 0.25        |
> |                    | weat4         | flan-t5-xl    | 0.15\*\*    | 0.28        |
> | P-AT-gender        | weat7         | flan-t5-xl    | 0.85\*\*    | 0.98        |
> |                    | weat8         | flan-t5-xl    | 0.6\*\*     | 0.83        |
> |                    | weat6         | flan-t5-xl    | 0.2\*\*     | 0.2         |
> | P-AT-age           | weat10        | flan-t5-xl    | 0.18      | 0.61        |
> | P-AT-base | weat1         | flan-t5-xxl   | 0.89\*\*    | 0.99        |
> |                    | weat2         | flan-t5-xxl   | 0.61\*\*    | 0.91        |
> |                    | weat3         | flan-t5-xxl   | 0.16\*\*    | 0.38        |
> |                    | weat3b        | flan-t5-xxl   | 0.0       | 0.0         |
> |                    | weat4         | flan-t5-xxl   | 0.12\*\*    | 0.31        |
> |                    | weat10        | flan-t5-xxl   | 0.05      | 0.21        |
> |                        | weat6         | flan-t5-xxl   | 0.1       | 0.22        |
> |                        | weat7         | flan-t5-xxl   | 0.65\*\*    | 0.8         |
> |                    | weat8         | flan-t5-xxl   | 0.55\*\*    | 0.78        |
> |                       | weat9         | flan-t5-xxl   | 0.31\*\*    | 0.93        |
> | P-AT-race          | weat3         | flan-t5-xxl   | 0.17\*\*    | 0.22        |
> |                    | weat4         | flan-t5-xxl   | 0.14\*     | 0.32        |
> | P-AT-gender        | weat7         | flan-t5-xxl   | 0.8\*\*     | 0.96        |
> |                    | weat8         | flan-t5-xxl   | 0.78\*\*    | 0.95        |
> |                    | weat6         | flan-t5-xxl   | 0.25\*\*    | 0.56        |
> | P-AT-age           | weat10        | flan-t5-xxl   | 0.4\*\*     | 0.88        |
>
> We absolutely agree that the entropy measure is useful, but it more specifically measures another aspect, related to language modeling abilities. However, we can observe that low entropy (data skewed toward a certain attribute) is often associated with a non-statistically significant difference in observed proportions, while the opposite is not true.
>
> **Addressed Issue: WEAT criticism, commenting reasons to reject**
>
> We chose to use WEAT as a starting point because of the links it has with bias in humans.
> It is certainly true that these new models may also need to be tested against more complex phenomena, yet our proposed tests also point out some of their biases.
> Regarding the metric problem, we agree that cancellation may occur.
> For this reason, we emphasize the need to observe a variety of subtasks and analyze each aspect in isolation. We kept the different tasks separate and decided to also report partial results on the different prompts, fixed a task. We aggregated only the results of semantically equivalent tests, which consist of sentences that are a rephrasing of the same basic question. We could emphasize more the equivalence of prompts in a P-AT task.
>
> **Addressed Issue: Order of attributes, answering second question**
>
> This is certainly an interesting problem. We have not tried this configuration, but we definitely plan to add it in a future extension of the present dataset.

---

### Meta-Review · Senior_Area_Chairs · 2023-10-04

**Recommendation:** 4

**Metareview:**

This paper presents an extension of the Word Embedding Association Test (WEAT) for Instruction-Following Language Models (IFLMs) like Alpaca and Flan-T5, referred to as P-AT. The study aims to detect biases across dimensions such as gender, race, and age in these models. In addition to the bias score, the paper introduces an "Entropy" metric to evaluate the quality of language modeling. The authors apply their bias assessment to various state-of-the-art models and discuss the relationship between bias and model sizes and the impact of prompt templates on bias. The reviewers' comments can be summarized as follows:
Reasons to Accept:
Valuable Tool: Reviewer 1 highlights the paper's contribution as a valuable tool for assessing bias within models that comprehend prompts. They appreciate the clear presentation, well-defined experiments, and the introduction of the Entropy metric.
Solid Support: Reviewer 3 finds the paper's identification of a gap in bias measurement for instruction-tuned models and the proposal of a novel metric well-motivated. They also acknowledge the solid evaluation, even though some presentation improvements are suggested.
Reasons to Reject:
Lack of Significance Testing: Reviewer 1 points out a key weakness in P-AT compared to previous work like SEAT/WEAT, which is the absence of direct measurement of statistical significance in observed bias. While entropy provides some evidence that the model is not ignoring prompts, it does not substitute for a significance test like the p-value.
Choice of WEAT: Reviewer 2 questions the choice of using WEAT as the basis for P-AT, suggesting that more recent and comprehensive bias detection datasets might be more appropriate. They also express concerns about the suitability of WEAT for probing bias in IFLMs.
Lack of Bias Definition: Reviewer 3 finds the paper lacking in providing concrete definitions for "bias" and "harm." They suggest linking the score to social issues and consider the brittleness of prompts in bias detection.

Overall Assessment:

The reviewers generally appreciate the paper's contribution to bias detection in IFLMs and the solid experimental support. However, concerns regarding the lack of significance testing, choice of WEAT, and the need for clearer definitions of bias and harm have been raised. Addressing these concerns could significantly strengthen the paper.

Based on the reviews, the paper demonstrates strong soundness and provides valuable contributions to the field, although it receives mixed excitement scores. The concerns raised by the reviewers, particularly regarding significance testing, choice of metrics, and clarity in definitions, should be addressed in a revised version to enhance the paper's overall quality.

---

### Decision · Program_Chairs · 2023-10-07

**Decision:**

Accept-Findings

**Comment:**

This paper presents an extension of the Word Embedding Association Test (WEAT) for Instruction-Following Language Models (IFLMs) like Alpaca and Flan-T5, referred to as P-AT. The study aims to detect biases across dimensions such as gender, race, and age in these models. In addition to the bias score, the paper introduces an "Entropy" metric to evaluate the quality of language modeling. The authors apply their bias assessment to various state-of-the-art models and discuss the relationship between bias and model sizes and the impact of prompt templates on bias. The reviewers' comments can be summarized as follows:
Reasons to Accept:
Valuable Tool: Reviewer 1 highlights the paper's contribution as a valuable tool for assessing bias within models that comprehend prompts. They appreciate the clear presentation, well-defined experiments, and the introduction of the Entropy metric.
Solid Support: Reviewer 3 finds the paper's identification of a gap in bias measurement for instruction-tuned models and the proposal of a novel metric well-motivated. They also acknowledge the solid evaluation, even though some presentation improvements are suggested.
Reasons to Reject:
Lack of Significance Testing: Reviewer 1 points out a key weakness in P-AT compared to previous work like SEAT/WEAT, which is the absence of direct measurement of statistical significance in observed bias. While entropy provides some evidence that the model is not ignoring prompts, it does not substitute for a significance test like the p-value.
Choice of WEAT: Reviewer 2 questions the choice of using WEAT as the basis for P-AT, suggesting that more recent and comprehensive bias detection datasets might be more appropriate. They also express concerns about the suitability of WEAT for probing bias in IFLMs.
Lack of Bias Definition: Reviewer 3 finds the paper lacking in providing concrete definitions for "bias" and "harm." They suggest linking the score to social issues and consider the brittleness of prompts in bias detection.

Overall Assessment:

The reviewers generally appreciate the paper's contribution to bias detection in IFLMs and the solid experimental support. However, concerns regarding the lack of significance testing, choice of WEAT, and the need for clearer definitions of bias and harm have been raised. Addressing these concerns could significantly strengthen the paper.

Based on the reviews, the paper demonstrates strong soundness and provides valuable contributions to the field, although it receives mixed excitement scores. The concerns raised by the reviewers, particularly regarding significance testing, choice of metrics, and clarity in definitions, should be addressed in a revised version to enhance the paper's overall quality.